# Global mean nitrogen recovery efficiency in croplands can be enhanced by optimal nutrient, crop and soil management practices

Luncheng You [1], Gerard H. Ros [2], Yongliang Chen [1] ✉, Qi Shao [1], Madaline D. Young [2], Fusuo Zhang [1] & Wim de Vries [2]

An increase in nitrogen (N) recovery efficiency, also denoted as N use efficiency (NUEr), is crucial to reconcile food production and environmental health. This study assessed the effects of nutrient, crop and soil management on NUEr accounting for its dependency on site conditions, including mean annual temperature and precipitation, soil organic carbon, clay and pH, by meta-regression models using 2436 pairs of observations from 407 primary studies. Nutrient management increased NUEr by 3.6-11%, crop management by 4.4–8%, while reduction in tillage had no significant impact. Site conditions strongly affected management induced changes in NUEr, highlighting their relevance for site-specific practices. Data driven models showed that the global mean NUEr can increase by 30%, from the current average of 48% to 78%, using optimal combinations of nutrient (27%), crop (6.6%) and soil (0.6%) management. This increase will in most cases allow to reconcile crop production with acceptable N losses to water. The predicted increase in NUEr was below average in most high-income regions but above average in middle-income regions.

Nitrogen (N) is the main limiting nutrient for photosynthetic processes and growth-development of crops[1]. Excessive use of N fertilizer leads to low N recovery efficiency, also denoted as N use efficiency (NUEr where r is added to indicate the link to recovery)[2], and elevated N losses to air and water with related impacts on terrestrial and aquatic ecosystems[3]. Global food production needs to increase by 50% to feed the world population projected for 2050[4], according to the World Resources Institute. For global food security and environmental benefits, there is an urgent need to implement optimal agricultural management strategies to further increase the current mean global NUEr (48%)[5,6].

Optimized agricultural management strategies increase NUEr and cover a combination of nutrient, crop, and soil management practices.

Nutrient management includes fertilizer strategies to increase NUEr by synchronizing crop demand and nutrient availability[7,8], using the right fertilizer type, with the right rate, at the right time, and at the right place[9–12]. Examples of the right fertilizer type include enhanced efficiency fertilizers[13–15] as well as smart combinations of inorganic and organic fertilizers[16,17]. Crop management can increase NUEr by exploiting differences in N uptake efficiencies between crop sequences[12,18], and includes diversity in crop rotations[11,19], use of cover crops[9,20] and recycling of crop residues[21,22]. In addition, soil management has often focused on tillage to reduce soil carbon decomposition[23,24], cultivation methods to enhance crop yields, soil biodiversity, and structural stability[25–28], and management of organic

[1]College of Resources and Environmental Sciences, National Academy of Agriculture Green Development, Key Laboratory of Plant–Soil Interactions, Ministry of Education, State Key Laboratory of Nutrient Use and Management, China Agricultural University, 100193 Beijing, China. [2]Wageningen University and Research, Environmental Systems Analysis Group, P.O. Box 47, 6700AA Wageningen, The Netherlands. ✉e-mail: ylchen@cau.edu.cn

residues to enhance soil nutrient levels to avoid nutrient deficiencies limiting crop growth[29].

In most cases, agricultural systems in high-income countries have higher NUEr than those in middle-income countries due to a more appropriate application of mineral fertilizers and mechanization[30]. The average cropland NUEr in the USA and European countries varies between 66 and 69%[31] whereas in China and India it varies between 21 and 35%[5,32]. This is due to stricter environmental and fertilizer regulations and the provision of reliable fertilizer recommendation systems in the USA and European countries as compared to China and India. In both China and India, there is an overapplication of relatively cheap urea with low efficiency due to high fertilizer subsidies and low urea prices. Consequently, country-specific fertilizer strategies are key to increasing NUEr in middle-income countries where excess N is applied. Conversely, the high NUEr of over 80% in sub-Saharan Africa is due to the reduced access to costly N fertilizers[33]. Farmers in low-income countries in sub-Saharan Africa thus have to increase the total nutrient input to increase crop yield whereas farmers in high-income countries need to expand the adoption of precision farming technologies and crop diversification strategies[2].

Agronomic practices can increase NUEr, but their impacts vary across regions due to variations in crop management, such as underuse of N in Africa[34] and overuse in China[35], and variations in local factors affecting the impact of N fertilizers, such as weather conditions and soil properties. Information on such variation is currently lacking at a global scale, leading to biased, and possibly unreliable predictions of the potential impact of agronomic practices to increase NUEr[36,37]. Recent meta-analyses have largely focused on assessing the impact of single agronomic measures on NUEr[13,15,23]. For example, Jiang et al.[15] found that enhanced efficiency fertilizer application increased rice NUEr by 20% globally compared to urea. The nature and degree of interactive impacts of management practices and site conditions on the actual NUEr are still not well understood, limiting a comprehensive assessment of these practices on NUEr. Therefore, there is a need to assess the impact of nutrient, crop, and soil management practices on NUEr while accounting for site conditions at a global scale.

In this study, we used meta-analytical and meta-regression models, to evaluate and predict the impacts of management practices on NUEr as a function of site conditions. We first developed a meta-model by combining existing meta-analytical studies ($n = 29$; Supplementary Table 1) to predict the change in NUEr in response to agronomic practices (Supplementary Table 2) and its dependency on site conditions, including mean annual temperature (MAT), mean annual precipitation (MAP), soil organic carbon (SOC), soil clay content and soil pH. We compared these outcomes to the results of a single meta-regression model (with change in relative, absolute, and standardized NUEr as response variable) based on 2,436 paired observations from 407 primary studies (Fig. 1, Supplementary Tables 3 and 4). We made this comparison to account for spatial variability in site properties and to quantify their interacting impacts on NUEr, to explore whether meta-regression models, using the original data underlying different meta-analytical studies, would give more insight in the change in NUEr in response to agronomic practices and in their variation as affected by site conditions. We finally evaluated the impact of management and site conditions controlling NUEr, and applied this model to predict the spatial variation of the potential impact of agronomic management practices on NUEr as a function of site conditions at a global scale.

## Results
### Impact of management practices on NUEr
Most of the agricultural management practices lead to positive responses in NUEr (Fig. 2a, Supplementary Table 5). Using insights from existing meta-analytical studies, 2 of 12 practices increased relative NUEr up to 38% on average, including right fertilizer placement (28%) and crop rotation (38%). When original experimental data were used for a meta-regression, 7 of 12 practices increased relative NUEr, including the use of efficient fertilizers (31%), combined mineral (16%), the right fertilizer placement (26%), rate (39%) and timing (24%), residue retention (24%) and cover cropping (22%). However, the results of both methods of analysis shows that the use of organic fertilizer, and zero or reduced tillage generally decreases NUEr by 1.2–9%.

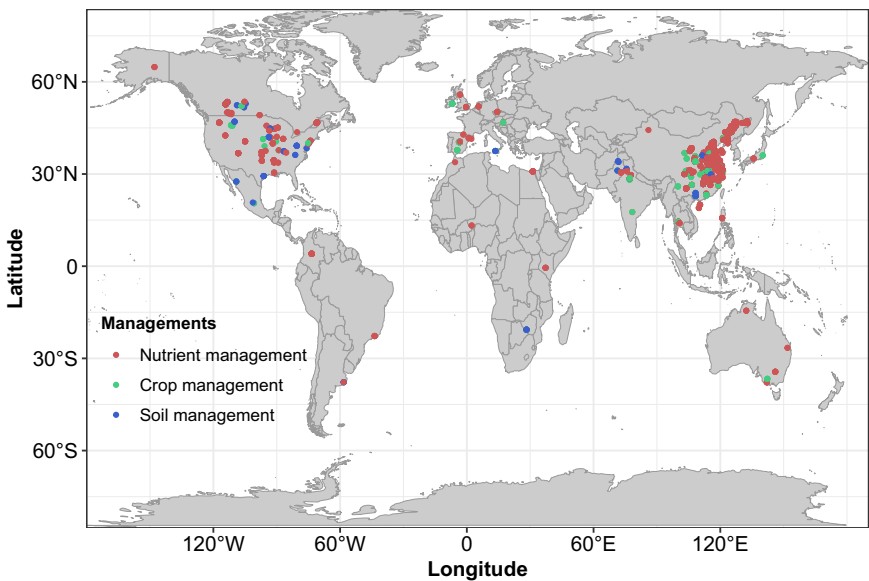

**Fig. 1 | World map indicating the locations of the 407 primary studies included in the study.** Sites are divided into experiments related to the impacts of nutrient management (enhanced efficiency fertilizer, combined fertilizer, organic fertilizer, right fertilizer placement, right fertilizer rate, and right fertilizer timing), crop management (residue retention, cover cropping, and crop rotation) and soil management (zero and reduced tillage). Source data are provided as a Source Data file.

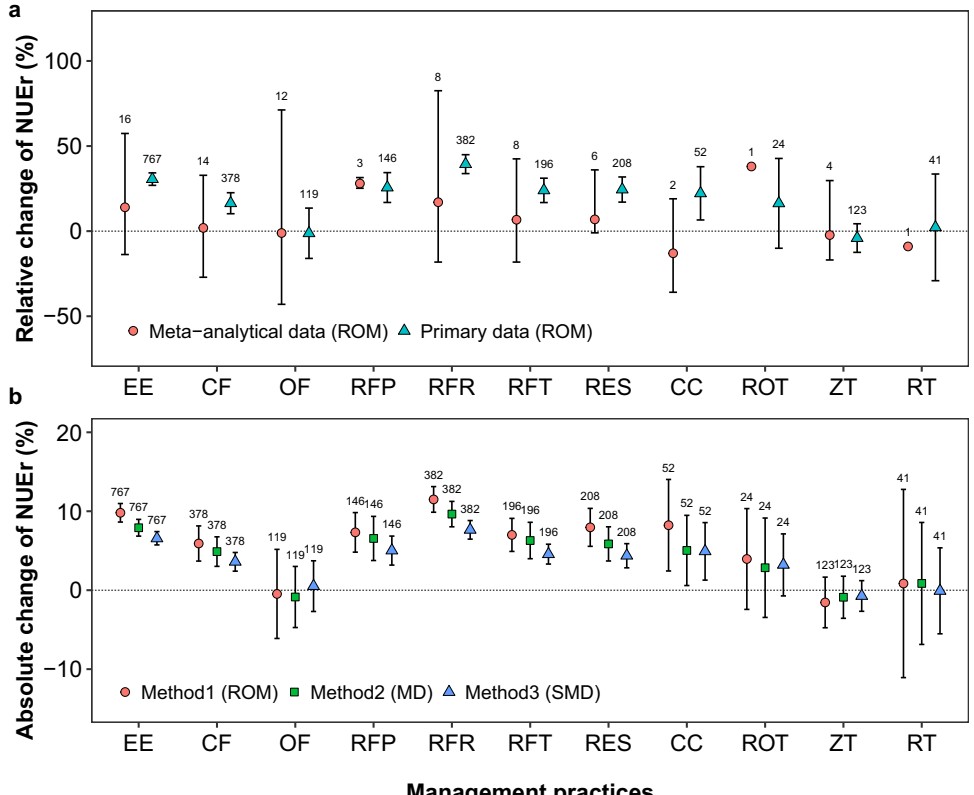

**Fig. 2 | Changes in N recovery efficiency (NUEr) in response to agricultural management practices based on the meta-analytical data and primary data of meta-analyses (see Supplementary Table 2).** **a** The relative change of NUEr, and **b** the absolute change of NUEr as compared to a control/treatment situation. Nutrient management includes enhanced efficiency fertilizer (EE), combined fertilizer (CF), organic fertilizer (OF), right fertilizer placement (RFP), right fertilizer rate (RFR), and right fertilizer timing (RFT). Crop management includes residue retention (RES), cover cropping (CC) and crop rotation (ROT). Soil management includes zero tillage (ZT) and reduced tillage (RT). ROM, the log-transformed ratio of means (Eq. 5); MD, the raw mean difference (Eq. 8); and SMD, the standardized mean difference (Eq. 10). Values above each box indicate the number of observations. Source data are provided as a Source Data file.

**Table 1 | The AIC values and corresponding analysis of variance $p$ values for the contribution of each variable to the N recovery efficiency (NUEr) computed with the main factor analysis with three models (ROM, MD, and SMD methods)**

| Variable | ROM method | | MD method | | SMD method | |
|---|---|---|---|---|---|---|
| | AIC | $p$ value | AIC | $p$ value | AIC | $p$ value |
| Fertilizer type | 78864 | <0.001 | 81114 | <0.001 | 9360 | <0.001 |
| Fertilizer strategy | 76452 | <0.001 | 79124 | <0.001 | 9363 | <0.001 |
| Residue retention | 81128 | 1 | 82535 | <0.01 | 9417 | 0.089 |
| Cover crop or crop rotation | 81117 | <0.01 | 82526 | <0.001 | 9416 | 0.055 |
| Zero or reduced tillage | 80996 | <0.001 | 82430 | <0.001 | 9352 | <0.001 |

Effect of moderators were tested using ANOVA using linear contrasts between model coefficients as implemented by metafor R package[63]. Source data are provided as a Source Data file.
*AIC* Akaike's information criteria, *ROM* the log-transformed ratio of means (Eq. 5); *MD* the raw mean difference (Eq. 8); *SMD* the standardized mean difference (Eq. 10).

When analyzing the change in absolute NUEr, there was a consistent increase in absolute NUEr up to 10% for most of the management practices applied, irrespective of the response variable used (ROM, MD or SMD) (Fig. 2b, Table 1 and Supplementary Table 6). The three response variables showed similar impacts of management on the NUEr but with lowest variance for absolute and standardized NUEr. Based on ROM, the absolute NUEr increased by applying enhanced efficiency fertilizer (9.8%) and combined fertilizer (5.9%), using the right fertilizer placement (7.3%), rate (11%) and timing (7%), as well as residue retention (8%) and cover cropping (8%), showing that the absolute NUEr can increase from 33–43% on average. In contrast, use of organic fertilizers, crop rotation, or zero or reduced tillage had limited effect on NUEr.

## Impacts of site conditions and management practice interactions on NUEr

The impact of management practices on NUEr was strongly affected by site conditions as shown by the meta-analytical studies (Supplementary Fig. 4) for categorial clusters of crop, soil and climatic conditions. We found a consistent positive impact of SOC, soil pH and MAP, and a negative impact of N application rate and clay content on NUEr when meta-regression models were calibrated on the original field observations (Fig. 3a–c, Supplementary Table 7). In all cases, the nutrient and crop management practices increased NUEr, while soil management showed the opposite impact and decreased NUEr. The effect of site conditions on the impacts of management on NUEr varied with the different practices as shown by the interaction between the right

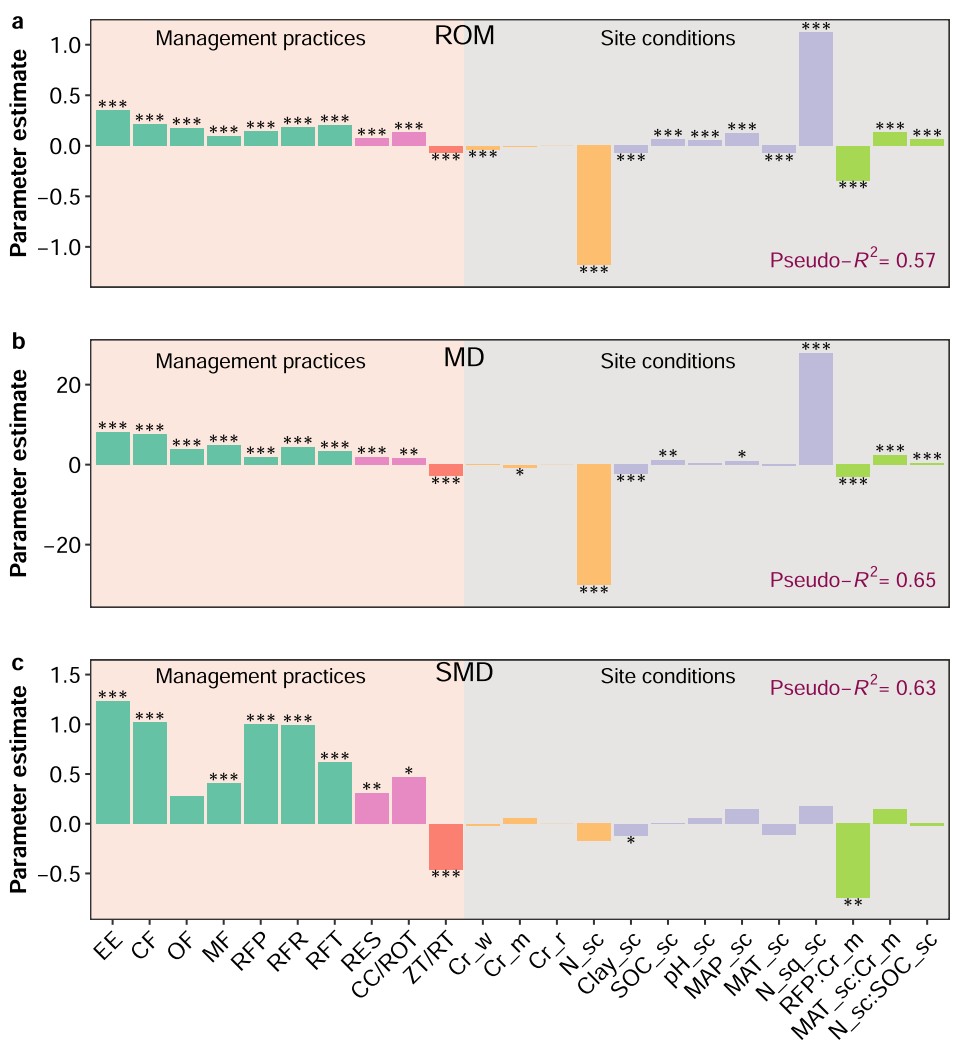

**Fig. 3 | Parameter estimates for the N recovery efficiency (NUEr) model. a** ROM method, **b** MD method, and **c** SMD method. The colon (:) signifies the interaction between variables. Asterisks indicate statistically significant differences (*$p < 0.05$, **$p < 0.01$ and ***$p < 0.001$). The effect of moderators was tested using ANOVA using linear contrasts between model coefficients as implemented by metafor R package[63]. Exact $p$ values are given in Source Data. The management practices include nutrient management (EE, enhanced efficiency fertilizer; CF, combined fertilizer; OF, organic fertilizer; MF, mineral fertilizer; RFP, right fertilizer placement; RFR, right fertilizer rate; and RFT, right fertilizer timing), crop management (RES, residue retention; CC, cover cropping; and ROT, crop rotation), soil

management (ZT/RT, zero or reduced tillage), and the site conditions include crop type, N rate, soil properties and climate (Cr_w, crop type wheat; Cr_m, crop type maize; Cr_r, crop type rice; N_sc, N application rate scaled; Clay_sc, soil clay content scaled; SOC_sc, soil organic carbon scaled; pH_sc, soil pH scaled; MAP_sc, mean annual precipitation scaled; MAT_sc, mean annual temperate scaled; and N_sq_sc, N application rate squared scaled). Scaled variables were converted to have unit variance. ROM, the log-transformed ratio of means (Eq. 5); MD, the raw mean difference (Eq. 8); SMD, the standardized mean difference (Eq. 10). Source data are provided as a Source Data file.

fertilizer placement and crop type (maize), and between N application rate and the organic carbon content, as well the interaction between MAP and crop type (maize) (Fig. 3). In addition, site conditions including crop type, soil pH, soil clay content, SOC, temperature and precipitation had all impact on the baseline NUEr.

The analysis of 2,436 paired observations from experiments from all over the globe showed that 57-65% of the variation in management-induced NUEr changes could be explained by the variation in site conditions (Fig. 3). The percentage of explained variance declined from the change in absolute NUEr (MD explained 65%; Fig. 3b) down to the change in relative NUEr (ROM explained 57%; Fig. 3a). The parameter estimates indicate that selecting the right fertilizer type or using a combination of organic and inorganic fertilizers increases the NUEr by 3.8-8.1% (Fig. 3b). Optimizing fertilization placement, rate and time further increase NUEr with 4.3%. NUEr was also increased by improved crop residue management (1.9%) and more diverse crop rotation (1.6%),

but decreased by zero or reduced tillage (-2.9%). As expected, higher N application rates decreased NUEr (-0.018% per kg of N added). Changes in NUEr varied by crop type (including wheat, maize, and rice), as well as clay content (-2.3%) and MAT (-0.4%) whereas the change in NUEr was positively correlated to MAP (1.0%), SOC (1.2%) and soil pH (0.3%).

## Global potential of optimizing cropping practices to increase NUEr

Given that the MD effect size is the most robust one in explaining the variation in NUEr in response to practices (Fig. 3), we used this regression model to predict the potential averaged effect of combined agronomic practices on absolute NUEr for all global croplands and its uncertainties (the lower and upper confidence limits for the NUEr changes) (Fig. 4), with impacts of combined nutrient, crop, and soil management practices in Fig. 5 and of each individual practice in Supplementary Fig. 5. The use of enhanced efficiency fertilizers,

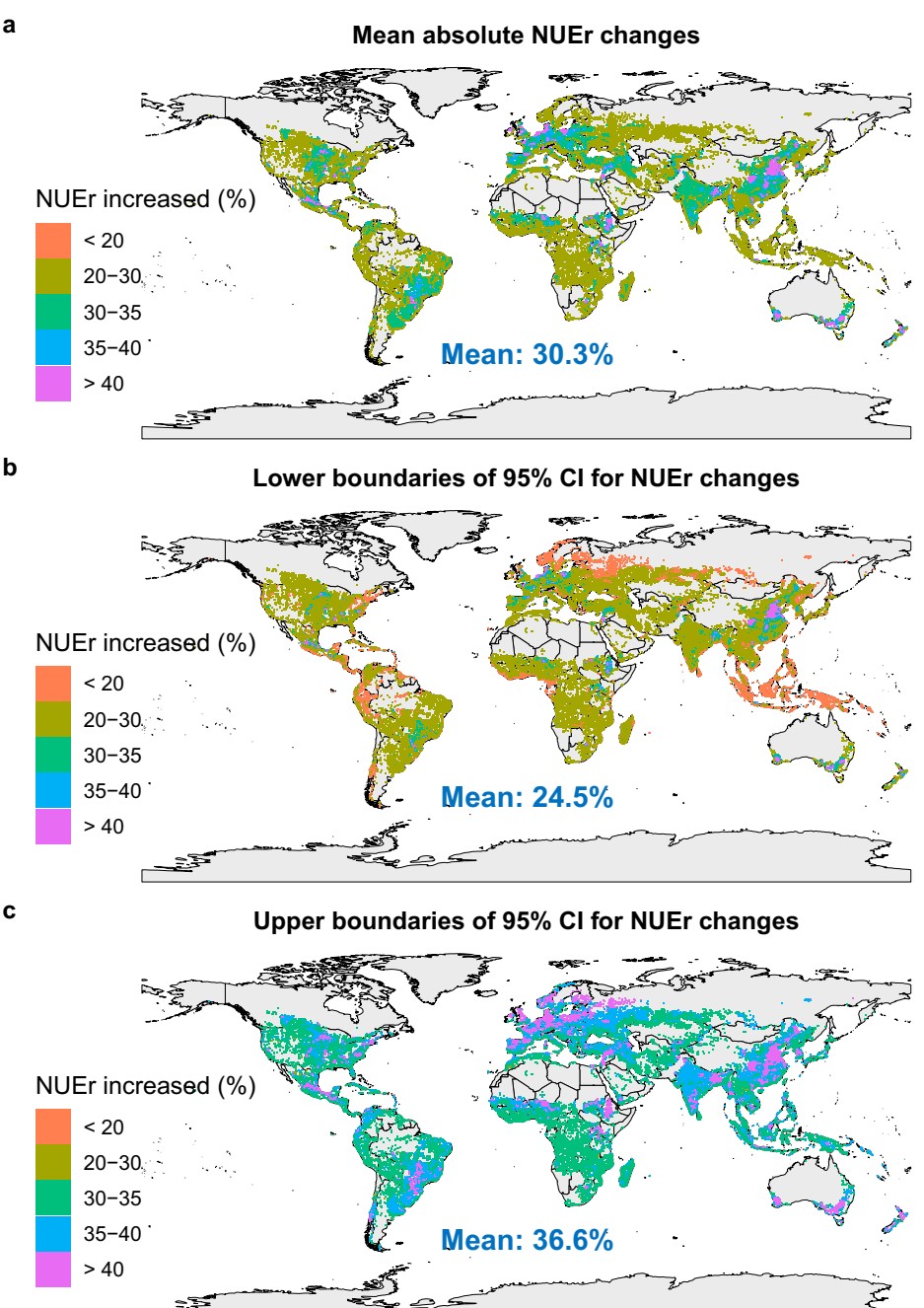

**Fig. 4 | Predicted spatial variation in impacts of combined optimal management practices on absolute average N recovery efficiency (NUEr) changes (%) in global croplands based on the MD (the raw mean difference) model. a** The mean of absolute average NUEr changes (%), **b** the lower and **c** upper boundaries of 95% confidence interval (CI) for NUEr changes (%). For an updated version of this figure taking into account the existing adoption of combined improved management practices in various regions, please see Figure 1 in our Addendum to this Article[74]. Source data are provided as a Source Data file.

combined fertilizer and organic fertilizer had an overall mean increase in NUEr of 6.4, 5.9 and 2.1% (Supplementary Fig. 5a–c) compared with a situation where only mineral fertilizer were applied. Optimal fertilizer placement, rate and timing increased mean NUEr by 4.7, 7.5, and 6.4% (Supplementary Fig. 5d–f). Combining these six management practices increased mean NUEr by 27% (Fig. 5a). Optimizing crop management via residue incorporation, cover crops, and crop rotation increased mean NUEr by up to 5% (Supplementary Fig. 5g, h), and combining these crop management practices resulted in increased mean NUEr of 6.6% (Fig. 5b). Zero or reduced tillage had little impact on mean NUEr (0.6%) (Fig. 5c). With the optimal

combination of nutrient, crop and soil management practices, global absolute NUEr could be increased by 30% on average (Fig. 4a). The uncertainties in the estimated increase of all combined agronomic practices were on average 6% (the mean of lower and upper boundaries was 25 and 37%, respectively). Optimal management practices had a higher impact on NUEr in eastern Africa, central Asia, southern North America, central South America, and Southern Australia compared to the global average. In contrast, optimal management practices have a relatively lower impact on NUEr in croplands in northern Europe, Southern Asia, and Eastern North America.

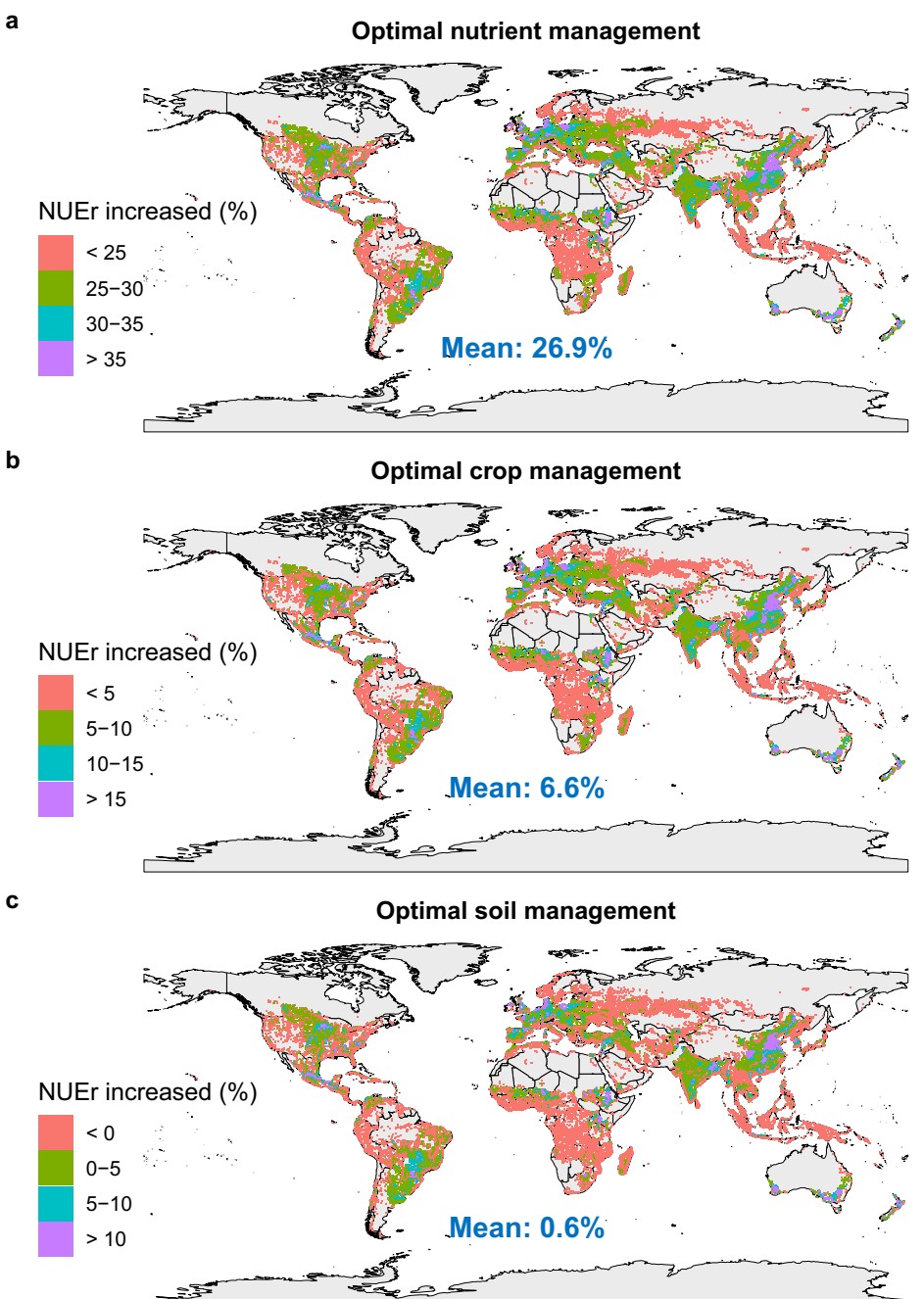

**Fig. 5 | Predicted spatial variation in impacts of management practices on absolute average N recovery efficiency (NUEr) changes (%) in global croplands based on the MD (the raw mean difference) model. a** Optimal nutrient management, **b** optimal crop management and **c** optimal soil management. Nutrient management included enhanced efficiency fertilizer (EE), combined fertilizer (CF), organic fertilizer (OF), right fertilizer placement (RFP), right fertilizer rate (RFR) and right fertilizer timing (RFT). Crop management included residue retention (RES), cover cropping (CC), and crop rotation (ROT). Soil management included zero tillage (ZT) and reduced tillage (RT). For an updated version of this figure taking into account the existing adoption of three individual improved management practices (nutrient, crop, and soil management) in various regions, please see Figure 2 in our Addendum to this Article[74]. Source data are provided as a Source Data file.

## Discussion

The results of this study reveal the potential impact of improved nutrient, crop, and soil management practices on NUEr as a function of site conditions. As expected, the fertilizer 4 R strategies had strong and positive effects on NUEr, as these practices ensure that crops receive adequate inputs for N during critical crop growth[2,38]. Applying the right fertilizer rate is an effective measure to reduce excess N volatilization into the air, runoff into adjacent lands and surface waters, or leaching to groundwater[39], since the N uptake per unit N applied decreases when N availability is not limiting crop growth. Right timing of fertilization (e.g., split application and weather-dependent application events) can improve the synchronization of the supply of applied N with crop requirements[7] thereby avoiding unnecessary losses, in particular in the beginning and final phase of crop growth. Right placement (e.g., fertilizer injection, fertilizer banding) can increase soil N concentration in the root zone and associated uptake rates[40], and reduce ammonia volatilization losses due to limited diffusion rates[41], in particular for urea or ammonia-based fertilizers. Similarly, higher NUEr values were observed after application of enhanced efficiency fertilizers, which can slow N transformation rates and result in the

minimization of particular loss paths prior to critical crop growth periods[42]. The positive effect of partial substitution of mineral fertilizers with organic fertilizers on NUEr agrees with field observations from long-term experiments[12,43,44] given the positive impacts of manure on the structure and nutrient retention capacity of soils. Organic manure additionally provides essential macro- and micronutrients in addition to N[44] and improves the soil microbial activity[45], thereby providing slow-release N in the later stages of crop growth[46]. Since only part of the manure N is directly available, NUEr often declines under full substitution, and a combination of both organic and mineral N is required to match crop demand and N supply throughout the growing season[43,47].

The literature indicates that integrated management of crop and soil not only increases NUEr but also improves the stability and resilience of agroecosystems to avoid growing conditions limiting crop yield[48]. For example, crop rotation improves the nutrient availability and water-holding capacity of the soil compared to monoculture[48,49]. Notably, cover cropping and residue retention were more important than crop rotation for improved NUEr, consistent with the results of studies[50,51] in which they found positive effects on soil organic matter and fertility, and crop yields. However, soil management practices, including zero or reduced tillage had limited impact on NUEr, with individual studies showing a small positive or negative effect, which is supported by other meta-analyses focusing on the yield stability in organic and conservation agriculture[27,40,52]. One possible explanation for the small negative impact in some studies is the more limited soil aeration when tillage is not practiced[53,54] causing a decrease in seedling emergence and crop production[55]. However, zero or reduced tillage is often proposed to be beneficial to a range of environmental variables compared to multiple-pass tillage, including soil biodiversity, organic matter, and soil structural stability[28]. Therefore, in order to benefit from those advantages of conservation tillage, while avoiding a decrease in NUEr, additional changes in farming practices are essential. For example, residue retention combined with crop rotation can minimize the negative impact of zero or reduced tillage on NUEr[27].

The effects of management practices on NUEr vary regionally depending on the N application rate, crop type, soil properties, and local climate. NUEr decreased with an increase in N application rate. This is mainly because N inputs exceeded the N requirements of the crop, which leads to excess N loss to water, soil, and air[2], and the current analysis focuses on agricultural systems that receive sufficient fertilizer inputs for optimum crop yields. Consequently, no experiments were included in which the soil was actively mined due to a higher N uptake than N input. For crop type, the response of NUEr to management practices varied greatly between cropping systems, such as lowland rice cropping systems and upland cropping systems, reflecting the variation in crop physiology as well as the associated management practices affecting N uptake. This difference in NUEr response to management practices between lowland and upland cropping systems may be related to soil aeration, which is poor in the lowland rice system due to water logging[56]. For soil properties, as expected, soil clay content had a negative effect on NUEr. The negative effect of clay content on NUEr was likely due to the low soil microporosity and poor gas exchange capacity with high clay content[57]. An increase in MAP had a positive effect on NUEr because lack of water affects crop growth and grain formation, especially in low rainfall regions, resulting in lower effective N uptake by the crop.

Optimizing nutrient and crop management practices can increase global NUEr by 27 and 6.6%, respectively. However, soil management practices have a limited effect on NUEr (close to 0). Considering a current NUEr of 48%[5,6], this implies a potential increase of up to 78%, and also a substantial reduction in N fertilizer inputs and reactive N

losses without negative impacts on food production and security. Given the variation in current management practices and site conditions, the impact of management on NUEr strongly differs across the continents. In most high-income regions, the increase in NUEr was below the average (32%), while NUEr increase was higher in middle-income regions such as central China, northern India, east Africa, southern North America, and central South America. This was mainly because the current NUEr (<40% on average) is lower in those regions[2], leading to a higher potential for improvement. Regions with low a NUEr due to excessive N inputs[58] (e.g., central China and northern India) benefit especially from applying an optimal N rate. In East African countries, including Kenya, Ethiopia, and Sudan[59], southern Mexico[60], and central Brazil[61], climate change risks (e.g., rising average temperatures and increasing precipitation extremes) limited NUEr in recent years, stressing the importance of management to mitigate these negative impacts due to climate change.

Assessing the global impact of management on NUEr from experiments being performed under highly variable conditions is challenging due to differences in methodology, regional and site conditions (often undefined), and limiting insights in the actual conditions affecting crop N uptake in the experiments done. Building a meta-model by combining individual meta-analyses (being calibrated on independent datasets) leads to higher uncertainties in predicted NUE than the meta-regression method that integrates data from individual experimental studies due to strong differences in methodology across meta-analyses and the confounding effect site conditions and management practices, factors that are known to interact for their impact on NUE. Differences in methodology are known to have substantial impacts on the derivation of aggregated effect sizes[62] due to differences in selection criteria, effect size indices, and the use of multivariate or multilevel methods to account for non-independent sampling errors or true effects. Regarding moderators, meta-analysis focuses on estimating an overall effect size, while meta-regression quantifies the impact of covariates on effect sizes. Where meta-analyses give valuable insights in the driving factors controlling NUE and might help to distinguish effective from ineffective measures to improve the NUE, their actual applicability on local, regional, and global scale in modelling studies or decision support systems seems rather limited. Strong differences between meta-analytical studies, as evaluated by Young et al.[52], have been confirmed by our analysis (Fig. 2), indicating that upscaling of empirical derived estimates for NUEr to regional or even global maps have to account for methodological differences in order to avoid skewed or even unreliable insights in the potential to improve NUEr. Meta-regression models can use different methods or response variables (effect sizes) to assess the factors controlling NUEr, but they differ in their ability to assess the impacts of practices. The effect sizes used in this study included ROM, MD, and SMD, and all of them confirmed the importance of site conditions and current management on the change in NUEr given the soil, crop, and nutrient factors evaluated, with MD and SMD providing more robust estimates (narrow confidence interval) than ROM. From standard error propagation rules, the uncertainty on a division is always greater than the uncertainty of a difference calculation[62] supporting the use of (S)MD above ROM. In addition, MD or SMD are easier to understand and interpret, because NUEr itself is already a response measure[63]. Given the existing data availability for site conditions, the MD was able to identify more interactions among site conditions and the observed change in NUEr than SMD, so we ultimately chose the MD model for upscaling.

By synthesizing available global meta-analyses and collecting the primary data from the underlying literature, we quantified the effects of nutrient, crop, and soil management practices on NUEr as a function of site conditions. Accounting for current management practices and site conditions, optimized agronomic practices can increase the global NUEr on cropland by 30% on average, with the highest impact on

nutrient management (27%), followed by crop management (6.6%) and soil management (0.6%). The largest increases are possible in countries that are currently struggling with N deficiencies as well as excess of N due to inappropriate fertilizer management. Considering a current global mean NUEr value of 48%, this implies that optimal agricultural management strategies may improve the mean global NUEr to nearly 80%. Increasing the global average NUEr up to this level, however, requires coordinated action to apply the most promising management practices while considering site conditions.

## Methods
### NUEr definition
Researchers have assigned different definitions for N use efficiency, thus requiring a clear definition when used. The two key approaches that are used to define and quantify N use efficiency include the N difference approach and the N balance approach[4]. In the N difference approach, generally used in agronomic studies, the N use efficiency is calculated as the difference in N uptake in total biomass (grain and crop residues) in a fertilized and unfertilized plot, divided by the fertilizer N input. This term is generally denoted as fertilizer N recovery efficiency. In the N balance approach, being the most widely used approach in environmental studies, the N use efficiency is calculated as the ratio of N harvested by crops divided by the total N input (including not only the N input by fertilizer but also other sources, i.e., N fixation and N deposition)[2]. In this study, we assessed the N use efficiency based on the N difference approach (the N recovery efficiency), since this is most relevant for agricultural practices. Also, the bulk of N use efficiency data collected in agronomic studies are based on an assessment of total aboveground plant N uptake in fertilized and unfertilized plots, while observations of N deposition and fixation, permitting calculation of total N input, are lacking. Some studies reporting only grain yield increase in response to added N fertilizer were not included. In our study the defined N use efficiency, denoted as NUEr to make the link to N fertilizer recovery, was thus calculated, according to Dobermann[64]:

$$NUEr = \frac{NUP_{fertilized} - NUP_{unfertilized}}{N_{rate}} \times 100, \qquad (1)$$

where, NUEr is expressed as a percentage (%), $NUP_{fertilized}$ and $NUP_{unfertilized}$ is the N uptake by aboveground plants (kg N ha$^{-1}$) in the fertilized treatment and unfertilized control during the experiment, respectively and $N_{rate}$ is the rate of N fertilizer applied (kg N ha$^{-1}$).

### Data collection
**Collection of meta-analytical studies.** In December 2021, we performed a literature search for meta-analytical studies on the effect sizes for NUEr or N uptake in response to changes in nutrient, crop and soil management. Searches were performed using Web of Science (https://www.webofscience.com) with search terms: NUEr, N uptake, nutrient management, crop management, soil management and meta-analysis (Supplementary Note 1). The meta-analytical studies included met three criteria: (1) linked to at least one management practice to the impact of NUEr or N uptake; (2) limited to management of main cereal croplands (maize, wheat and rice), excluding grasslands and forests; and (3) providing estimates based on field studies, thus excluding laboratory or incubation studies. When meta-analytical studies presented a summary of previous analyses, only the most recent study was selected. This search and selection resulted in the inclusion of 29 studies (Supplementary Fig. 1). Detailed information about these studies is given in Supplementary Data 1, including bibliographic details, crop types, management practices and response variables, with a summary in Supplementary Table 1. Supplementary Table 1 describes the management methods and controls, divided over (1) nutrient management: enhanced efficiency fertilizer, combined fertilizer, organic

fertilizer, mineral fertilizer, fertilizer placement, fertilizer rate and fertilizer timing; (2) crop management: residue retention, cover cropping and crop rotation; and (3) soil management including zero and reduced tillage. For each management practice, the control (treatment) situation is mentioned to which the practice is compared is given in Supplementary Table 2.

**Collection of the primary data.** Relevant nutrient, crop and soil management data and site conditions were retrieved from the 407 primary studies based on the 29 meta-analytical studies. This resulted in 2436 paired observations for maize, wheat and rice (Supplementary Data 2). From these studies the following variables were extracted: (1) reference details including author, title and publication year; (2) latitude and longitude; (3) experiment duration; (4) site-specific soil properties and climatic conditions; (5) crop type; (6) number of replicates; (7) management practices applied (in predefined nutrient, crop and soil management classes); (8) mean NUEr in experimental and control treatments; and (9) practices of variation (including standard error, 95% confidence interval or standard deviation). When replicate numbers were not reported, the number of replicates of the primary studies was estimated as 3. If studies did not provide standard deviations (SD) or standard errors, the SD was estimated from the mean coefficient of variation (CV) of other studies in the database[65] as:

$$SD_{NUEr,i} = CV_{NUEr} \times NUE_{r,i} \times 1.25, \qquad (2)$$

where, $CV_{NUEr}$ is the mean CV of NUEr values provided.

In most of the primary studies, information on site conditions that might have affected the impacts of practices, i.e., climate and soil properties, was lacking. To be consistent, all those data were derived from the given longitude and latitude, using climate data from CRU (Climate Research Unit) database (http://www.cru.uea.ac.uk/data), i.e., MAT and MAP; and soil properties from Soil Grids (http://www.isric.org/explore/soilgrids), i.e., clay content, SOC and soil pH.

An overview of the data collected is given in Supplementary Table 3. The sampling sites and locations of the 407 primary studies are shown in Fig. 1. The distribution is shown in Supplementary Fig. 2. Most of the study sites were located in Asia (77%), North America (14%), and Europe (4%), and less in South America (2%), Africa (2%) and Australia (1%) (Supplementary Data 2). Maize, wheat, and rice accounted for 35, 30, and 35% of the total primary studies, respectively. The most evaluated management practices were enhanced efficiency fertilizers (31%), combined fertilizer (15%), and fertilizer rate (15%) followed by crop residue (9%), fertilizer timing (8%), fertilizer placement (6%), zero tillage (6%), organic fertilizer (5%), cover cropping (2%), reduced tillage (2%) and crop rotation (1%). An overview of the NUEr mean, variance, and range for the control and treated plots, and the variation in site conditions is given in Supplementary Table 4. The mean NUEr of experimental treatments (39%) was 6% higher than the mean of control treatments (33%). The majority of the studies had NUEr values ranging between 20 and 60% for the control plots in which no additional practices had been taken to increase NUEr. About one-fifth of the NUEr values were under 20%, and only one-tenth of the NUEr values was over 60% (Supplementary Data 2). Site conditions of the analyzed studies cover the main range of variability in global agricultural regions, with MAT ranging from -0.6 to 29 °C, MAP from 45 to 2330 mm, soil organic carbon content from 2.7 to 80 g kg$^{-1}$, soil pH from 4.5 to 8.5 and clay content from 8.8 to 53%.

### Data analysis
**Meta-model integrating the published meta-analytical studies.** Multiple observations or treatments were collected in meta-analytical studies, which means that data points were correlated.

When the same management practices were reported by multiple meta-analyses, the overall mean change in NUEr due to the measure and the associated standard error were calculated by the following equations to establish one meta-model from the assessed meta-analytical studies[52].

$$\bar{x} = \frac{\sum(x_i/\sigma_i^2)}{\sum(1/\sigma_i^2)} \qquad (3)$$

and

$$\sigma_{\bar{x}} = \frac{1}{\sqrt{\sum(1/\sigma_i^2)}}, \qquad (4)$$

where, $\bar{x}$ is the weighted mean, $\sigma_{\bar{x}}$ is the standard error of weighted mean, $x_i$ is the individual mean from the effect size reported, and $\sigma_i^2$ is the individual variance from the effect size reported.

**Assessing mean effects of practices on NUEr derived from original field studies.** In order to conduct a meta-regression on the original experimental data derived from the 407 primary studies, we first calculated the effect sizes and corresponding variances of the primary studies using three methods (also called effect sizes) based on the means, standard deviations and number of repetitions of the recorded NUEr values[63,66,67].

The log-transformed ratio of means (ROM) was calculated as:

$$\ln RR = \ln\left(\frac{X_t}{X_c}\right), \qquad (5)$$

where, $X_t$ and $X_c$ are the mean NUEr in the treatment and control groups, respectively.

The corresponding variance was calculated as:

$$V_{\ln RR} = \frac{s_t^2}{n_t X_t^2} + \frac{s_c^2}{n_c X_c^2} \qquad (6)$$

where, $n_t$ and $n_c$ are the number of the treatment and control, respectively, and $s_t$ and $s_c$ are the standard deviations of the treatment and control, respectively. The change in relative NUEr (as %) compared to the control due to a management measure was subsequently calculated as:

$$\text{Relative change (\%)} = \left(e^{\ln RR} - 1\right) \times 100 \qquad (7)$$

The raw mean difference (MD) was calculated as:

$$MD = X_t - X_c \qquad (8)$$

The corresponding variance was calculated as:

$$V_{MD} = \frac{s_t^2}{n_t} + \frac{s_c^2}{n_c} \qquad (9)$$

The standardized mean difference (SMD) was calculated as:

$$SMD = \frac{(X_t - X_c)}{SD_p} \qquad (10)$$

and

$$SD_p = \sqrt{\frac{(n_t-1)s_t^2 + (n_c-1)s_c^2}{n_t + n_c - 2}} \qquad (11)$$

where, $SD_p$ is the pooled within-group standard deviation.

The corresponding variance was calculated as:

$$V_{SMD} = \frac{n_t + n_c}{n_t \times n_c} + \frac{SMD^2}{2(n_t + n_c)} \qquad (12)$$

Given that the collected data came from studies applying different research methods, there is non-independence and heterogeneity among the effects[62,65]. We accounted for the non-independence by using multivariate meta-modeling with restricted maximum-likelihood estimation, as implemented in Metafor[65]. Paper number was used to specify the random-effects structure of the model. NUEr observations of the primary studies were assumed to be independent whereas effects within each study received correlated random effects assuming a symmetric compound structure. Random-effects models can estimate the distribution of individual effect sizes of means, residual heterogeneity and sampling error[65]. It calculates the mean effect size as a weighted mean of individual effect sizes, using the inverse of the sum of the between-study variance (due to variation in experimental conditions) and within-study variance (due to sampling error) as weights[66].

To compare the results of ROM, MD, and SMD methods, all results are expressed as changes in absolute NUEr. For the ROM method, the average NUEr in the control group ($\bar{X}_c$) for different management practices was calculated firstly, and then the absolute NUEr was calculated based on the relative NUEr:

$$Absolute\ change_{(ROM)}(\%) = Relative\ change(\%) \times \bar{X}_c \qquad (13)$$

For the SDM method, the average pooled within-group standard deviation ($\overline{SD_p}$) for different management practices was initially calculated, and then the absolute NUEr was calculated as:

$$Absolute\ change_{(SMD)}(\%) = SMD \times \overline{SD_p} \qquad (14)$$

**Assessing impact of site conditions controlling NUEr from original field studies**

To evaluate the impact of management practices and site conditions (MAP, MAP, clay content, SOC, and soil pH) on NUEr derived from original field studies, a main factor analysis was performed initially to assess their overall impact. The principle behind this approach is based on generalized conclusions derived from a large number of field studies, allowing the identification of broadly applicable cause-effect relationships. An analysis of variance was then done to evaluate the contribution of each of the assessed management practices and site conditions on the variation of the NUEr[65], combined with an analysis of both Akaike's information criteria (AIC) and the $p$ value.

Since the impact of nutrient, crop, and soil management practices on NUEr may interact, we analyzed the main and all two-way interactions between management practices and site conditions using a mixed effects model with interaction terms[62]:

$$y_i = \beta_0 + \beta_1 x_{i1} + \beta_2 x_{i2} + \beta_3 x_{i1} x_{i2} + \dots + u_i + e_i \qquad (15)$$

where, $y_i$ is the observed effect size of NUEr, $x_{i1}$ is the value of the first moderator variable for the $i$th study and $x_{i2}$ is the value of the second moderator variables for the $i$th study, $\beta_0$ is a regression coefficient representing the intercept, $\beta_1$ is a regression coefficient indicating how the average true effect size changes for one unit increase in $x_{i1}$, $\beta_2$ is a regression coefficient indicating how the average true effect size changes for one unit increase in $x_{i2}$, $u_i$ is the variance of the true effect (residual heterogeneity) of study i, $e_i$ is the sampling error of study i, and $x_{i1}x_{i2}$ is the interaction term with coefficient $\beta_3$.

To avoid overfitting the regression model, we first checked for unacceptably high predictive correlations before fitting the model (Supplementary Fig. 3). We assessed the impact of each factor and its

interaction with other variables using analysis of variance. Pseudo-$R^2$ values (McFadden's method) and AIC were used to compare regression models. The best model had high pseudo-$R^2$ and low AIC values. We also checked the amount of residual heterogeneity according to the $Q_E$ output of the *rma.mv* function in R 4.2.2 software[65]. $Q_E$ tests show whether the variability in the observed effect size (for which the moderators do not account) is larger than the expected sampling variability only, so $Q_E$ represents the heterogeneity that cannot be explained by the model. Smaller values for $Q_E$ reflect a better model performance.

**Assessing spatial variation and global potential to increase NUEr and its uncertainties.** The model developed to assess changes in NUEr due to agronomic practices as a function of site conditions (Eq. 15) was used to make a spatial explicit assessment of the impacts of those measures by applying the derived empirical model to all croplands around the globe using global data sets on site conditions. We thus estimated the global potential for NUEr improvements on a 0.5 ×0.5 degree resolution using existing global data sets of: (1) N inputs by fertilizer and manure from PANGAEA[68] (Data Publisher for Earth & Environmental Science) database (https://doi.org/10.1594/PANGAEA.871980), (2) climate data from CRU (http://www.cru.uea.ac.uk/data), including MAT and MAP, (3) land use data from the SPAM (Spatial Production Allocation Model) dataset[69] (https://www.mapspam.info/data); and (4) soil properties from Soil Grids (http://www.isric.org/explore/soilgrids) including clay content, SOC and soil pH. We mapped both the average NUEr increase and the associated uncertainties, expressed by accounting for the variance of the effect of studies (Eq. 15) while neglecting the uncertainty in site data (N inputs, climate data, land use data, and soil properties). The uncertainty in site data can locally be large but levels out at the coarse 0.5 ×0.5 degree resolution as being used in this study. Uncertainties in predicted NUEr change were given by calculating the 95% confidence interval around the predicted change in the mean NUEr, being constructed based on the critical values from a standard normal distribution (i.e., 1.96 for 95%) where the predicted values are based only on the fixed effects (betas from Eq. 15) of the model[63,70]. In the analysis, we assumed that management practices across global croplands are entirely conventional, without explicitly accounting for the existing adoption of improved agronomic measures in various regions. This assumption may have led to an overestimation of the potential for enhancing nitrogen recovery efficiency (NUEr) globally. More specifically, we did not account for several aspects that have now been considered in an updated analysis published in an Addendum to this Article[74]: (1) Regional variability in management practices. The analysis did not incorporate detailed maps or datasets reflecting the current adoption rates of improved practices, such as enhanced efficiency fertilizers, optimized fertilizer placement, cover cropping, and conservation tillage. (2) Technology-level differences. We used the SPAM dataset to classify cropland into low, high, and intermediate technology levels, but did not link implementation of conventional versus enhanced practices to the technology level as we did in the revised estimate. In the Addendum[74] we now also admit the limitation of the assumption that enhanced efficiency fertilizers are applied in high technology areas. (3) Crop residue and tillage practices. The analysis did not account for regional variations in crop residue management (e.g., retention vs. burning) or the adoption of conservation tillage practices. (4) Fertilizer use efficiency. Adjustments to fertilizer rates were not explicitly linked to regional differences in nitrogen use efficiency (NUE), particularly in areas with low NUE (<0.5).

**Other uncertainties in the impacts of measures on the spatial variation in NUEr increase.** The 407 primary studies included in this study were conducted at the plot scale to quantify the NUEr in the soil-plant system. For crop farms, the NUEr at plot scale can be considered representative for the farm scale, assuming that the current agricultural management practices at the experimental locations are equal

to the traditional management practices of the farmers. Also, while the NUEr definition at plot scale and farm scale differs for a livestock farmer, it is similar for a crop farmer, with N inputs and N outputs from soils belonging to a farm being equal to N fertilizer inputs and crop N outputs to and from the farm[71].

Since the variations in NUEr changes at plot scale are based on the local climate conditions, soil properties, and traditional agronomic practices, the results can also be used to predict the variation at a plot (farm) scale and NUEr at a global scale. However, there are significant uncertainties in this upscaling due to the unevenly distributed data sets (the data set in this study is mainly concentrated in USA and China, while other regions are relatively scarce) that we used in assessing the impacts of site properties on management impacts and the uncertainties in global data sets on N inputs by fertilizer and manure, climate data, land use data, and soil properties. Additional studies are needed to assess the impact of management practices on the NUEr in the Animal-Plant-Soil System and in the Agro-Food system, in view of N losses in the crop-animal system (from feed to animal products) and in the total food chain[72]. The latter information is also needed to support policies and actions for sustainable agricultural management.

## Reporting summary

Further information on research design is available in the Nature Portfolio Reporting Summary linked to this article.

## Data availability

N inputs by fertilizer and manure is available from: PANGAEA (Data Publisher for Earth & Environmental Science) database (https://doi.org/10.1594/PANGAEA.871980). Climate data is available from CRU (http://www.cru.uea.ac.uk/data). Land use data is available from the SPAM (Spatial Production Allocation Model) dataset (https://www.mapspam.info/data). Soil property data is available from Soil Grids (http://www.isric.org/explore/soilgrids). The raw data are available on GitHub[73] at https://github.com/gerardhros/phd_luncheng/tree/main/articles/ncoms23. Source data are provided with this paper.

## Code availability

The code is available on GitHub[73] at https://github.com/gerardhros/phd_luncheng/tree/main/articles/ncoms23.

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

## Acknowledgements

Y.L.C. was supported by the National Key Research and Development Program of China (2021YFD1700900). L.C.Y. acknowledges funding from the China Scholarship Council (No.201913043) and Hainan University.

## Author contributions

L.C.Y., G.H.R., Y.L.C., and W.D.V. conceived and designed the study. Data collection fieldwork was led and supervised by G.H.R, and W.D.V. Data was collected by L.C.Y. and Q.S. The analysis was conducted by L.C.Y., G.H.R., and M.D.Y. Figures were produced by L.C.Y. with help from G.H.R. Results were critically assessed and interpreted by L.C.Y., G.H.R., Y.L.C., F.S.Z., and W.D.V. The manuscript draft was written by L.C.Y., G.H.R., Y.L.C., and W.D.V. All authors edited and approved the manuscript.

## Competing interests

The authors declare no competing interests.
