## [Peer Review File · Nature Communications]

Reviewers' Comments:

Reviewer #1:

Remarks to the Author:

The main new contribution of this paper is that it provides sound and systematically analyzed field evidence for significant improvement in NUE, i.e. realistic but substantial increases can be achieved through fertilizer management and better agronomic practices. That provides hope and it underpins targets that have often been stated (e.g. 70%), but without demonstrating how they can be achieved. Overall, I found the paper well-written and clear in its main results and conclusions and would like to see it published after some revision. I have the following suggestions for that:

1) It took me a while to realize that the definition of NUE used in this analysis is different from the NUE used in other (global) analyses, including Zhang et al (2015). Eq, 1 (L334) indicates that what you have calculated here is in fact the recovery efficiency (in the crop) of applied N fertilizer, which is not the same as NUE calculated from a typical N output/N input approach. That must be clarified upfront.

2) Following this, you cannot refer to papers such as Zhang et al (2015) when talking about increases in NUE to be achieved. Zhang's global average was 42% for cropland in 2010, but based on the input-output budget, not fertilizer recovery. Likewise, the global average NUE in the new FAO cropland budget (<https://www.fao.org/faostat/en/#data/ESB>) was 55% in 2020, but also based on the N budget. The latter also indicates that NUE may have already risen quite a bit in the most recent 10 years.

3) Considering the above, I think the discussion -- also of the influencing factors -- needs to mainly refer to a number of other papers that have tried to review NUE in the recovery efficiency sense, for example: Cassman et al. <https://pubmed.ncbi.nlm.nih.gov/12078002/>, Ladha et al. <https://www.sciencedirect.com/science/article/pii/S0065211305870038>, or similar ones. Irrespective of that, 70% is still a good global target for NUE, either defined as recovery efficiency or defined as N outputs/N inputs.

4) The spatial extrapolation to the global scale is problematic, for two reasons: (i) the data set is biased towards few world regions (Fig. 4) such as N. America and China, whereas others (e.g. E. Europe, SE Asia, Africa, much of S.America) are hardly present; (ii) the grid data sources from PANGAE are in itself associated with large uncertainties. There is not much you can do about that, but I suggest adding a discussion on that to the paper, pointing out some limitations of this study.

L29 and other occurrences throughout: it appears that the 40% NUE value is sourced from Zhang et al (2015). They reported 42% in 2010, but used a different NUE calculation. It may well be that the average global recovery efficiency of applied N is also now about 40% (see Cassman et al 2002), but we don't really have a global database for that.

L39: These are old FAO food demand projections from 2012. Look for something newer. It'll probably be more like 50% increase, not 60%.

L42: see previous comment on definition and baseline of NUE

L49: cited here are two papers on biochar, which is a rather niche solution and also not an organic fertilizer of the common kind. Cite manure!

L61-62: other reasons include: production quota, subsidies and use of cheap ammonium bicarbonate with low efficiency by farmers in China for a long time, small fields (=easy to over-apply), fertilizer subsidies/urea price control in India (no incentive to do better), environmental and fertilizer regulations (N. America, Europe).

L76: zero tillage is very rare in rice

L101, 104: I cannot see how biochar can ever become a scalable technology.

L195: that is true in most crops but not all (e.g. not in lowland rice or plantation crops).

L205: There is much literature available now showing that zero tillage is not a major measure to reduce GWP. that has been grossly overestimated before, particularly with regard to the soil C sequestration potential.

Reviewer #2:

Remarks to the Author:

General

The manuscript focusses on the role of management (nutrient, crop and soil) on NUE. This is a relevant topic that has been underestimated on the scientific literature and so it is a major original contribution to the state of the art. Actually, the results emphasize that nutrient and crop management can increase global NUE by 30%, much more than other approaches that are deserving attention in the scientific literature. The work will be of significance for the fields of Agronomy and Environmental Sciences. Therefore, I believe that the manuscript present novel and relevant results. Nevertheless, some major clarifications are needed before publication to strength the study.

Results and Discussion

In the list of meta-analysis (Supplementary, Table 1) all the studies are focus mainly on grain cereals except for one focus on vegetable crops and several that say 'other crops'. Later on the manuscript all comments refer only to the main cereal crops (maize, wheat and rice). Therefore, two questions arise: 1) how different would be the results if the analysis were conducted only with the main cereal crops (maize, wheat and rice); 2) maybe the manuscript should focus only on the main cereals and leave aside the rest of crops. Particularly, vegetables are very different from arable crops (i.e. N output differs greatly from N uptake in fruit vegetables) and might be adding extra noise to the results.

Differences in the results provided by the meta-analytical models and the meta-regression with original data (Fig. 2) should be discussed further. It is confusing for the reader to understand why they are different and which one are more trustful. A short explanation emphasizing the common conclusions derived from both approaches and the uncertainties associated with the divergent results could be helpful.

The effect of SOC on NUE should be discussed further. From all the environmental variables, SOC has the stronger relationship with NUE and it is negatively related (Supplementary Fig.2). This result need and explanation. It suggests that lower NUE is expected in fertile soils, probably because it is complicated to estimate the optimal N rate as the N supply by soil mineralization is relevant for crop uptake but also hard to predict. Please, discuss.

In relation with the previous comment, the sentence (L.125-126) '...while soil management showed the opposite impact and decreased NUE' needs to be further discussed. The positive effect of nutrient and crop management on NUE is a solid and logical result from this study. The fact that soil management is having negative or no-effect on NUE is disturbing and needs to be clarify. Even more, are they negative interactions between nutrient and soil management? Please, discuss further.

Material and Methods

There is a need to clearly define NUE. From the text (L.281-288) and equation 1 (L. 334) it seems that NUE was defined as the fraction of N applied that was take up by the crop. However, this definition corresponds to N recovery efficiency (Lhada et al. 2005, *Advances in Agronomy*). The most common NUE definition is N output/Ninput, where the N output is the exported out of the cropping system (Lassaletta et al., 2014, *Environmental Research Letters*; Zhang et al., 2015, *Nature*). The authors need to address this issue and clarify if they use data of N recovery efficiency and NUE together, and if they did what are the implications of pooling together these two different variables (Quan et al. 2021, *Nature Food*).

There is a need to clarify the system scale at which NUE was calculated. This issue is related to the NUE definition and are both very relevant. Different NUE quantification approaches are usually related to the system scale (Zhang et al., 2020, *Global Biogeochemical cycles*). In the current

study soil-plant scale was chosen to calculate NUE. The authors should justify why and how these results could be compared with farm scale studies (i.e. Quemada et al. 2020, Agricultural Systems).

Specific comments

L. 201-202 'One possible explanation for this impact is the limited soil aeration causing seedling emergence and crop production'. It seems that a word is lacking in this sentence, maybe the authors mean '...causing a decrease in seedling emergence and crop production'.

L. 207 Remove brackets from '(changes in)'

L. 228 In this discussion, I suggest that the authors clearly state the difference and say 'Optimizing nutrient and crop management practices can increase NUE by...'. Later, they can add 'Additionally, soil management practices can increase/decrease/have no effect on NUE...'

Responses to reviewers' comments to NCOMMS-23-12278 "The global mean nitrogen use efficiency in croplands can be enhanced to 70% by optimal nutrient, crop and soil management practices"

We greatly appreciate the insightful comments and suggestions to our manuscript by the reviewers. We revised our manuscript according to those comments and suggestion. The updates refer specifically to: (i) a description of the definition of NUE used for this study, (ii) the scale of the study and why and how the results at plot scale can be compared with farm scale studies for crop farms and (iii) limiting the focus of the study to the main cereal crops (maize, wheat and rice) only. In view of the last point, we thus removed all other crops, thereby reducing the analysis' noise, and we updated all the related figures, tables and corresponding text. Please find below our point-by-point responses and associated revisions in the new version.

All responses have been marked in blue, and our revisions have been highlighted in red in the main text.

Responses to Reviewer #1 (Remarks to the Author):

[Comment 1] The main new contribution of this paper is that it provides sound and systematically analyzed field evidence for significant improvement in NUE, i.e. realistic but substantial increases can be achieved through fertilizer management and better agronomic practices. That provides hope and it underpins targets that have often been stated (e.g.70%), but without demonstrating how they can be achieved. Overall, I found the paper well-written and clear in its main results and conclusions and would like to see it published after some revision. I have the following suggestions for that:

Response: Thanks for the appreciation and see further responses below. Note that the title of our paper even changes to nearly 80% since we now consistently use the term N recovery efficiency, also denoted as N use efficiency (NUE). This is explained in more detail below.

[Comment 2] It took me a while to realize that the definition of NUE used in this analysis is different from the NUE used in other (global) analyses, including Zhang et al (2015). Eq, 1 (L334)

indicates that what you have calculated here is in fact the recovery efficiency (in the crop) of applied N fertilizer, which is not the same as NUE calculated from a typical N output/N input approach. That must be clarified upfront.

Response: Thank you for your suggestions. In order to make our paper clearer, we have defined NUE in the “Materials and methods” section, but already shortly refer to it in “Abstract” and “Introduction”, respectively. The various added texts are listed as follows:

Abstract: “An increase in nitrogen (N) recovery efficiency, also denoted as N use efficiency (NUE), is ...”

Introduction: “Excessive use of N fertilizer leads to low N recovery efficiency, also denoted as N use efficiency (NUE)² ...”

Materials and methods: “Researchers have assigned different definitions for nitrogen use efficiency (NUE), thus requiring a clear definition when used. The two key different approaches that are used to define and quantify NUE include the N difference approach and the N balance approach⁴. In the N difference approach, generally used in agronomic studies, the NUE is calculated as the difference in N uptake in total biomass (grain and crop residues) in a fertilized and unfertilized plot, divided by the fertilizer N input. This term is generally denoted as fertilizer N recovery efficiency. In the N balance approach, being the most widely used approach in environmental studies, the NUE is calculated as the ratio of N harvested by crops divided by the total N input (including not only the N input by fertilizer but also other sources, i.e., N fixation and N deposition)³³. In this study, we assessed the N recovery efficiency, further denoted as NUE, since this is most relevant for agricultural practices. Furthermore, the bulk of NUE data collected in agronomic studies are based on an assessment of total aboveground plant N uptake in fertilized and unfertilized plots, while observations of N deposition and fixation, permitting calculation of total N input, are lacking. Few studies reporting only grain yield increase in response to added N fertilizer were not included. In our study the defined NUE was thus calculated, according to ⁶⁴:

$$NUE = \frac{NUP_{fertilized} - NUP_{unfertilized}}{N_{rate}} \times 100, \quad (1)$$

where, NUE is expressed as a percentage (%), $NUP_{fertilized}$ and $NUP_{unfertilized}$ is the N uptake by aboveground plants (kg N ha^{-1}) in the fertilized treatment and unfertilized control during the experiment, respectively and N_{rate} is the rate of N fertilizer applied (kg ha^{-1}).”

Comment 3] Following this, you cannot refer to papers such as Zhang et al (2015) when talking about increases in NUE to be achieved. Zhang's global average was 42% for cropland in 2010, but based on the input-output budget, not fertilizer recovery. Likewise, the global average NUE in the new FAO cropland budget (<https://www.fao.org/faostat/en/#data/ESB>) was 55% in 2020, but also based on the N budget. The latter also indicates that NUE may have already risen quite a bit in the most recent 10 years.

Response: Thanks. We agree that we caused confusion here by mixing the NUE derived by an N difference approach and an N balance approach, as defined above. We now updated the text by relating it to the global nitrogen recovery efficiency (simply denoted as NUE), which equals 48% according to two recently published papers (Yan et al., 2020; Quan et al., 2021). Thus, we cited the two papers and updated the NUE from 40% to 48% in the revised manuscript.

References:

- 5 Quan, Z., Zhang, X., Fang, Y. & Davidson, E. A. Different quantification approaches for nitrogen use efficiency lead to divergent estimates with varying advantages. *Nat. Food* **2**, 241–245 (2021).
- 6 Yan, M., Pan, G., Lavallee, J. M. & Conant, R. T. Rethinking sources of nitrogen to cereal crops. *Glob. Change Biol.* **26**, 191–199 (2020).

[Comment 4] Considering the above, I think the discussion -- also of the influencing factors -- needs to mainly refer to a number of other papers that have tried to review NUE in the recovery efficiency sense, for example: Cassman et al. <https://pubmed.ncbi.nlm.nih.gov/12078002/>, Ladha et al. <https://www.sciencedirect.com/science/article/pii/S0065211305870038>, or similar ones. Irrespective of that, 70% is still a good global target for NUE, either defined as recovery efficiency or defined as N outputs/N inputs.

Response: Thanks. Following your suggestions, we have added the two above mentioned relevant references, i.e., Quan et al., (2021) and Yan et al., (2020).

[Comment 5] The spatial extrapolation to the global scale is problematic, for two reasons: (i) the data set is biased towards few world regions (Fig. 4) such as N. America and China, whereas others

(e.g. E. Europe, SE Asia, Africa, much of S.America) are hardly present; (ii) the grid data sources from PANGAE are in itself associated with large uncertainties. There is not much you can do about that, but I suggest adding a discussion on that to the paper, pointing out some limitations of this study.

Response: We agree with the concern. According to your suggestions, we have added a short discussion about uncertainty and limitations in the section of spatial extrapolation to the global scale in the “Materials and methods”. The content is as follows:

“However, there are significant uncertainties in this upscaling due to the unevenly distributed data sets (the data set in this study is mainly concentrated in USA and China, while other regions are relatively scarce) that we used in assessing the impacts of site properties on management impacts and the uncertainties in global data sets on N inputs by fertilizer and manure, climate data, land use data and soil properties. Additional studies are needed to assess the impact of management practices on the NUE in the Animal-Plant-Soil System and in the Agro - Food system, in view of N losses in the crop - animal system (from feed to animal products) and in the total food chain⁷¹. The latter information is also needed to support policies and actions for sustainable agricultural management.”

Reference:

71 Zhang, X. et al. Quantifying nutrient budgets for sustainable nutrient management. *Global Biogeochem. Cycles* **34**, e2018GB006060 (2020).

[Comment 6] L29 and other occurrences throughout: it appears that the 40% NUE value is sourced from Zhang et al (2015). They reported 42% in 2010, but used a different NUE calculation. It may well be that the average global recovery efficiency of applied N is also now about 40% (see Cassman et al 2002), but we don't really have a global database for that.

Response: We agree with the concern. As stated above, the reported current global NUE (nitrogen recovery efficiency) is about 48%, and we used this new number in the revised manuscript, and also cited two references (Yan et al., 2020; Quan et al., 2021) as mentioned above.

[Comment 7] L39: These are old FAO food demand projections from 2012. Look for something newer. It'll probably be more like 50% increase, not 60%.

Response: Following your suggestions, we have changed “60%” to “50%” according to the new report below.

Reference:

4 Searchinger, T. et al. Creating a Sustainable Food Future: a Menu of Solutions to Feed Nearly 10 Billion People by 2050 (*World Resources Institute*, 2019).

[Comment 8] L42: see previous comment on definition and baseline of NUE

Response: According to your previous comments, we change this sentence to “For global food security and environmental benefits, there is an urgent need to implement optimal agricultural management strategies to further increase the current mean global NUE (48%)^{5,6}”.

[Comment 9] L49: cited here are two papers on biochar, which is a rather niche solution and also not an organic fertilizer of the common kind. Cite manure!

Response: Following your suggestions, we used two new references replacing the two biochar papers. More specifically we removed:

“14 Biederman, L. A. & Harpole, W. S. Biochar and its effects on plant productivity and nutrient cycling: a meta-analysis. *Global Change Biology Bioenergy* **5**, 202-214 (2013).

15 Dai, Y., Zheng, H., Jiang, Z. & Xing, B. Combined effects of biochar properties and soil conditions on plant growth: A meta-analysis. *Science of the Total Environment* **713**, 136635 (2020).”

And we added instead:

“16 Du, Y., Cui, B., Zhang, Q., Wang, Z., Sun, J. & Niu, W. Effects of manure fertilizer on crop yield and soil properties in China: A meta-analysis. *Catena* **193**, 104617 (2020).

17 Wei, Z., Ying, H., Guo, X., Zhuang, M., Cui, Z. & Zhang, F. Substitution of mineral fertilizer with organic fertilizer in maize systems: A meta-analysis of reduced nitrogen and carbon emissions. *Agronomy* **10**, 1149 (2020).”

[Comment 10] L61-62: other reasons include: production quota, subsidies and use of cheap ammonium bicarbonate with low efficiency by farmers in China for a long time, small fields (=easy to over-apply), fertilizer subsidies/urea price control in India (no incentive to do better), environmental and fertilizer regulations (N. America, Europe).

Response: Thanks for your suggestion. We have changed the sentence

“This is partly due to socioeconomic factors and the provision of reliable fertilizer recommendation systems.”

to

“This is due to stricter environmental and fertilizer regulations and the provision of reliable fertilizer recommendation systems in the USA and European countries as compared to China and India. In China, agriculture is dominated by small fields prone to overapplication of cheap ammonium bicarbonate with low efficiency, and India lacks proper fertilizer subsidies and urea price controls.”

[Comment 11] L76: zero tillage is very rare in rice.

Response: Thanks for your suggestion. We have changed the sentence

“For example, Liang et al.²¹ found by meta-analysis that zero tillage reduced rice NUE by 17% globally compared to multiple-pass tillage practices.”

to

“For example, Jiang et al.¹⁵ found that enhanced efficiency fertilizer application increased rice NUE by 20% globally compared to urea.”

[Comment 12] L101, 104: I cannot see how biochar can ever become a scalable technology.

Response: We agree with the concern. According to the suggestion of reviewer 2, we only retained the main cereal crops (maize, wheat and rice), and removed the other crops to reduce the statistical noise in the data. Since biochar management measure was mainly applied in the removed meta-

analytical studies and primary studies, the results of biochar addition on NUE are no longer included in the revised manuscript.

[Comment 13] L195: that is true in most crops but not all (e.g. not in lowland rice or plantation crops).

Response: We agree with the concern. Crops are concretized according to references 48 and 49: Maize and other dryland crops in rotation (such as maize-soy rotation, maize-wheat-soy rotation) versus maize monoculture. We rewrote the sentence “For example, crop rotation improves the nutrient availability and water holding capacity of the soil compared to monoculture^{48,49}.”

References:

- 48 Bowles, T. M. et al. Long-term evidence shows that crop-rotation diversification increases agricultural resilience to adverse growing conditions in north america. *One Earth* **2**, 284-293 (2020).
- 49 Tiemann, L. K., Grandy, A. S., Atkinson, E. E., Marin-Spiotta, E. & McDaniel, M. D. Crop rotational diversity enhances belowground communities and functions in an agroecosystem. *Ecol. Lett.* **18**, 761-771(2015).

[Comment 14] L205: There is much literature available now showing that zero tillage is not a major measure to reduce GWP. that has been grossly overestimated before, particularly with regard to the soil C sequestration potential.

Response: We agree with you. To avoid confusion to the potential readers, we removed this sentence in the revised manuscript.

Responses to Reviewer #2 (Remarks to the Author):

[Comment 1] General. The manuscript focusses on the role of management (nutrient, crop and soil) on NUE. This is a relevant topic that has been underestimated on the scientific literature and so it is a major original contribution to the state of the art. Actually, the results emphasize that nutrient and crop management can increase global NUE by 30%, much more than other approaches that are deserving attention in the scientific literature. The work will be of significance for the fields of Agronomy and Environmental Sciences. Therefore, I believe that the manuscript present novel and relevant results. Nevertheless, some major clarifications are needed before publication to strength the study.

Response: We greatly appreciate your suggestions. We have revised our manuscript based on your comments. Please check our point-by-point responses below.

[Comment 2] Results and Discussion. In the list of meta-analysis (Supplementary, Table 1) all the studies focus mainly on grain cereals except for one focus on vegetable crops and several that say ‘other crops’. Later on the manuscript all comments refer only to the main cereal crops (maize, wheat and rice). Therefore, two questions arise: 1) how different would be the results if the analysis were conducted only with the main cereal crops (maize, wheat and rice); 2) maybe the manuscript should focus only on the main cereals and leave aside the rest of crops. Particularly, vegetables are very different from arable crops (i.e. N output differs greatly from N uptake in fruit vegetables) and might be adding extra noise to the results.

Response: We agree with the concern. We thus only retained the main cereal crops (maize, wheat and rice), and removed the meta-analytical studies and the underlying data from primary studies of vegetables and other crops. Since vegetable and other crops data represent a small proportion of the total database (less than 5% of the total data), there is little impact on the overall results. We reanalyzed all the data and updated all the figures, tables, and related texts in the revised manuscript. We found that the trend of results did not change before and after deleting this part of the data.

[Comment 3] Differences in the results provided by the meta-analytical models and the meta-regression with original data (Fig. 2) should be discussed further. It is confusing for the reader to understand why they are different and which one are more trustful. A short explanation emphasizing the common conclusions derived from both approaches and the uncertainties associated with the divergent results could be helpful.

Response: We expected that meta-regression models, using the original data underlying different meta-analytical studies, would give more insight in the change in NUE in response to agronomic practices and in their variation as affected by site factors than the meta-analytical models. We thus made this comparison to see if this was true. Existing meta-regression models often reduce the variation in site properties by using categorized variables such as soil and climate properties as well as local management practices. In addition, these existing models have been developed with a specific focus (analyzing the impact of one or two measures while ignoring the others) and often show differences in methodology. So, combining existing meta-regression models into a single meta-model will evidently confound our possibilities to unravel the impact of site properties on the NUE. We now added the following text in the revised manuscript:

“We made this comparison to account for spatial variability in site properties and to quantify their interacting impacts on NUE, to explore whether meta-regression models, using the original data underlying different meta-analytical studies, would give more insight in the change in NUE in response to agronomic practices and in their variation as affected by site factors.”

[Comment 4] The effect of SOC on NUE should be discussed further. From all the environmental variables, SOC has the stronger relationship with NUE and it is negatively related (Supplementary Fig.2). This result needs an explanation. It suggests that lower NUE is expected in fertile soils, probably because it is complicated to estimate the optimal N rate as the N supply by soil mineralization is relevant for crop uptake but also hard to predict. Please, discuss.

Response: Thanks, but it was a mistake. Figure S2 (now included as Fig. S3) does not show the correlation between SOC and NUE, but showed the negative correlations between SOC and MAT, soil pH, and N rate, and also the positive correlations between SOC, MAP and clay content. In fact,

SOC is positively correlated with NUE (Fig. 3), implying that a higher NUE is found in fertile soils, according to expectations.

[Comment 5] In relation with the previous comment, the sentence (L.125-126) ‘...while soil management showed the opposite impact and decreased NUE’ needs to be further discussed. The positive effect of nutrient and crop management on NUE is a solid and logical result from this study. The fact that soil management is having negative or no-effect on NUE is disturbing and needs to be clarified. Even more, are there negative interactions between nutrient and soil management? Please, discuss further.

Response: We did so in the discussion by adding that one possible explanation for this impact is the more limited soil aeration when tillage is not practiced^{53,54} causing a decrease in seedling emergence and crop production⁵⁵.

[Comment 6] Material and Methods. There is a need to clearly define NUE. From the text (L.281-288) and equation 1 (L. 334) it seems that NUE was defined as the fraction of N applied that was taken up by the crop. However, this definition corresponds to N recovery efficiency (Lhada et al. 2005, *Advances in Agronomy*). The most common NUE definition is $N_{\text{output}}/N_{\text{input}}$, where the N output is the exported out of the cropping system (Lassaletta et al., 2014, *Environmental Research Letters*; Zhang et al., 2015, *Nature*). The authors need to address this issue and clarify if they use data of N recovery efficiency and NUE together, and if they did what are the implications of pooling together these two different variables (Quan et al. 2021, *Nature Food*).

Response: We greatly appreciate your suggestions. First of all, we added a section in the methods describing our definition of NUE, which is indeed N recovery efficiency (see added section in methods and the extensive reply to a similar question by reviewer 1).

We re-checked the meta-analysis database and the primary database, and found that the meta-analytical studies mainly used N recovery efficiency (recovery efficiency of fertilizer N based on total aboveground plant N). To ensure comparability of results, we only retained the data defined by N recovery efficiency (efficiency of total aboveground plant N), and removed the NUEs defined by other terms (less than 5% of the total data). We found that the trend of results did not change before and after deleting this part of the data.

[Comment 7] There is a need to clarify the system scale at which NUE was calculated. This issue is related to the NUE definition and are both very relevant. Different NUE quantification approaches are usually related to the system scale (Zhang et al., 2020, Global Biogeochemical cycles). In the current study soil-plant scale was chosen to calculate NUE. The authors should justify why and how these results could be compared with farm scale studies (i.e. Quemada et al. 2020, Agricultural Systems).

Response: Thanks for the suggestions. We first added a short text explaining that the plot scale used is representative for the farm scale in case of crop farms. We then added a short text on the different system scales and spatial scales that can be used when calculating the NUE.

Regarding the linkage between plot scale and farm scale, we added this text in the revised manuscript: “The 407 primary studies included in this study were conducted at the plot scale to quantify the NUE in the soil-plant system. For crop farms, the NUE at plot scale can be considered representative for the farm scale, assuming that the current agricultural management practices at the experimental locations are equal to the traditional management practices of the farmers. Furthermore, while the NUE definition at plot scale and farm scale differs for a livestock farmer, it is similar for a crop farmer, with N inputs and N outputs from soils belonging to a farm being equal to N fertilizer inputs and crop N outputs to and from the farm⁷⁰.”

Regarding different NUE quantification approaches related to different system scales, we added this text in the revised manuscript “Additional studies are needed to assess the impact of management practices on the NUE in the Animal-Plant-Soil System and in the Agro-Food system, in view of N losses in the crop-animal system (from feed to animal products) and in the total food chain⁷¹. The latter information is also needed to support policies and actions for sustainable agricultural management.”

References

- 70 Leip, A., Britz, W., Weiss, F. & de Vries, W. Farm, land, and soil nitrogen budgets for agriculture in Europe calculated with CAPRI. *Environ. Pollut.* **159**, 3243-3253 (2011).
- 71 Zhang, X., Davidson, E. A., Zou, T., Lassaletta, L., Quan, Z., Li, T. & Zhang, W. Quantifying nutrient budgets for sustainable nutrient management. *Global Biogeochem.*

[Comment 8] *Specific comments* L. 201-202 ‘One possible explanation for this impact is the limited soil aeration causing seedling emergence and crop production’. It seems that a word is lacking in this sentence, maybe the authors mean ‘...causing a decrease in seedling emergence and crop production’.

Response: Thanks for pointing out this. We have changed the sentence

“One possible explanation for this impact is the limited soil aeration^{52,53} causing seedling emergence and crop production⁵⁴.”

to

“One possible explanation for the small negative impact in some studies is the more limited soil aeration when tillage is not practiced^{53,54} causing a decrease in seedling emergence and crop production⁵⁵.”

[Comment 9] L. 207 Remove brackets from ‘(changes in)’

Response: Revised as suggested. We have removed the brackets.

[Comment 10] L. 228 In this discussion, I suggest that the authors clearly state the difference and say ‘Optimizing nutrient and crop management practices can increase NUE by...’. Later, they can add ‘Additionally, soil management practices can increase/decrease/have no effect on NUE...’

Response: Thanks for this suggestion. We have changed the sentence

“Optimizing nutrient, crop and soil management practices can increase global NUE by 30% on average.”

to

“Optimizing nutrient and crop management practices can increase global NUE by 27 and 6.6%. However, soil management practices have limited effect on NUE (close to 0).”

Reviewers' Comments:

Reviewer #1:

Remarks to the Author:

I am satisfied with the responses to my earlier comments and the revisions made in the paper. I have, however, a few final suggestions:

- To avoid confusion, change the title to "The global nitrogen recovery efficiency in croplands can be enhanced to nearly 80% by...."
- To make it crystal-clear throughout, consider using the acronym NUEr (or REN) instead of NUE
- L65: ammonium bicarbonate was used much in the past in China, but not so much anymore today
- L66: India does not 'lack' fertilizer subsidies and price controls: to the contrary, it has them and that is the reason for over-application of fertilizer

The authors have added additional comments in L483-495 to address problems in the extrapolation to the global scale. Nevertheless, given the available spatial data for the extrapolation as well as the substantial spatial data gaps in the field data that were used for regression model development, the uncertainty in the resulting spatial predictions is likely to be large. Hence, the paper would benefit from having that uncertainty quantified and mapped as well, which I think is feasible to suitable statistical simulation techniques. The result of that analysis should also be discussed in the main manuscript, to provide sufficient context for the predictions made.

Reviewer #2:

Remarks to the Author:

Overall, the authors answered to all queries from the reviewer and the revised version of the manuscript incorporates most of the suggestions. However, I still have a question with respect to NUE, given that in the revised version it is clear that the authors are analyzing the N recovery efficiency I believe that they should keep that clear in the manuscript. Particularly:

- The title should be 'The global mean nitrogen recovery efficiency in croplands can be enhanced to nearly 80% by optimal nutrient, crop and soil management practices'
- Material and Methods: L.294-295: The sentence 'In this study, we assessed the N recovery efficiency, further denoted as NUE, since this is most relevant for agricultural practices' should be rephrased as it is confusing and clearly state the focus of the study. It could be replaced by something like: 'In this study, we assessed the N recovery efficiency since this is most relevant for agricultural practices'

L.295-298: Think about modifying the sentence. In many studies the unfertilized plot is an indirect, but quite solid, measurement of N deposition and fixation.

Responses to reviewers' comments to NCOMMS-23-12278A "The global mean nitrogen use efficiency in croplands can be enhanced to nearly 80% by optimal nutrient, crop and soil management practices"

We appreciate the last comments and suggestions to our revised version of the manuscript by the two reviewers. We revised our manuscript again according to those comments and suggestions. The update refers specifically to an added quantification of the uncertainty in the spatial predictions. Please find below our point-by-point responses and associated revisions in the main text. All responses have been marked in blue, and our revisions have been highlighted in red in the main text.

Responses to Reviewer #1 (Remarks to the Author):

[Comment 1] I am satisfied with the responses to my earlier comments and the revisions made in the paper.

Response: Thanks for the appreciation. We have revised our manuscript based on your comments. Please check our point-by-point responses below.

[Comment 2] I have, however, a few final suggestions:

- To avoid confusion, change the title to "The global nitrogen recovery efficiency in croplands can be enhanced to nearly 80% by...."

Response: We now did so but still kept the use of NUE but adding an r (as suggested below) to indicate the link of the defined NUE with N recovery, while referring to definitions given by Zhang et al (2015).

Comment 3] To make it crystal-clear throughout, consider using the acronym NUER (or REN) instead of NUE.

Response: Thank you for your suggestion. We now made this change, using the acronym NUER instead of NUE.

[Comment 4] L65: ammonium bicarbonate was used much in the past in China, but not so much anymore today.

Response: Thanks. You are right. Urea is the dominant N fertilizer which has a relatively low NUE due to high NH₃ emissions, although being used more than ammonium bicarbonate today in China. We have changed the text now to “*relatively cheap urea*” and combined the reason for overapplication in both China and India (see response to comment 5).

[Comment 5] L66: India does not 'lack' fertilizer subsidies and price controls: to the contrary, it has them and that is the reason for over-application of fertilizer.

Response: We agree with the concern. According to your suggestions, we have changed the text to: “*In both China and India, there is overapplication of relatively cheap urea with low nitrogen efficiency due to high fertilizer subsidies and low urea prices.*”

[Comment 6] The authors have added additional comments in L483-495 to address problems in the extrapolation to the global scale. Nevertheless, given the available spatial data for the extrapolation as well as the substantial spatial data gaps in the field data that were used for regression model development, the uncertainty in the resulting spatial predictions is likely to be large. Hence, the paper would benefit from having that uncertainty quantified and mapped as well, which I think is feasible to suitable statistical simulation techniques. The result of that analysis should also be discussed in the main manuscript, to provide sufficient context for the predictions made.

Response: We agree with this concern, and we now quantified the uncertainty in the spatial predictions by calculating the 95% confidence interval, and mapped both the average NUEr increase and the associated uncertainties (Fig. 4).

In addition, we have added the following text to the “*Materials and methods*” section: “*We mapped both the average NUEr increase and the associated uncertainties, expressed by accounting for the variance of the effect of studies (see Eq. 15) while neglecting the uncertainty in site data (N-inputs, climate data, land use data and soil properties). The uncertainty in site data can locally be large*”

but levels out at the coarse 0.5 x 0.5 degree resolution as being used in this study. Uncertainties in predicted NUEr change were given by calculating the 95% confidence interval around the predicted change in the mean NUEr, being constructed based on the critical values from a standard normal distribution (i.e., 1.96 for 95%) where the predicted values are based only on the fixed effects (the betas from Eq. 15) of the model^{63,70}.”

70 refers to an added reference, in which a similar approach was used, i.e.

Schulte - Uebbing, L. F., Ros, G. H., & de Vries, W. Experimental evidence shows minor contribution of nitrogen deposition to global forest carbon sequestration. *Glob. Chang. Biol.* 28, 899-917 (2022).

Finally, we add the following text to the "Result" section:

“...we used this regression model to predict the potential averaged effect of combined agronomic practices on absolute NUEr for all global croplands and its uncertainties (the lower and upper confidence limits for the NUEr changes) (Fig. 4), with impacts of combined nutrient, crop and soil practices in Fig. 5 and of each individual practice in Supplementary Fig. 5.”

and

“The uncertainties in the estimated increase of all combined agronomic practices were on average 6% (the mean of lower and upper boundaries was 25% and 37%, respectively).”

Responses to Reviewer #2 (Remarks to the Author):

[Comment 1] Overall, the authors answered to all queries from the reviewer and the revised version of the manuscript incorporates most of the suggestions.

Response: We greatly appreciate your appreciation and suggestions. We have revised our manuscript based on your comments. Please check our point-by-point responses below.

[Comment 2] However, I still have a question with respect to NUE, given that in the revised version it is clear that the authors are analyzing the N recovery efficiency I believe that they should keep that clear in the manuscript. Particularly:

- The title should be 'The global mean nitrogen recovery efficiency in croplands can be enhanced to nearly 80% by optimal nutrient, crop and soil management practices'.

Response: Thank you for your suggestions. We changed the title to: *“The global mean nitrogen recovery efficiency in croplands can be enhanced to nearly 80% by optimal nutrient, crop and soil management practices”*.

[Comment 3] Material and Methods: L.294-295: The sentence 'In this study, we assessed the N recovery efficiency, further denoted as NUE, since this is most relevant for agricultural practices' should be rephased as it is confusing and clearly state the focus of the study. It could be replaced by something like: 'In this study, we assessed the N recovery efficiency since this is most relevant for agricultural practices'.

Response: Thanks. Following your suggestions, we have changed it to *“In this study, we assessed the N use efficiency based on the N difference approach (the N recovery efficiency), since this is most relevant for agricultural practices.”* to still make the link to the rest of the paragraph where N use efficiency is used.

[Comment 4] L.295-298: Think about modifying the sentence. In many studies the unfertilized plot is an indirect, but quite solid, measurement of N deposition and fixation.

Response: Thanks. We agree that this statement holds when a system is not fertilized for a long period, but this is not the case when the unfertilized plot has been fertilized in the past and part of the N uptake comes from mineralized N from the soil. We have made an assessment of the N uptake in non-fertilized plots but in most cases, this was much higher than N deposition and expected N fixation, indicating that many non-fertilized plots used in the experiments have been fertilized in the recent past.

Finally, we like to mention that we found mistakes in the description of the Eqs 6, 9 and 11 in the “*Materials and methods*” part which are now also updated (indicted by a red color).

Reviewers' Comments:

Reviewer #1:

Remarks to the Author:

I have no further comments. My suggestions have been addressed.

Reviewer #2:

Remarks to the Author:

I am satisfied with the revisions made in the manuscript.

Given the uncertainties associated to the calculations, the title could better reflect the results by removing 'to nearly 80%'.

Responses to reviewers' comments to NCOMMS-23-12278B "The global mean nitrogen recovery efficiency in croplands can be enhanced to nearly 80% by optimal nutrient, crop and soil management practices "

We appreciate the last comments and suggestions to our revised version of the manuscript by the two reviewers. We revised our manuscript again according to those comments and suggestions. Please find below our point-by-point responses and associated revisions in the main text. All responses have been marked in blue, and our revisions have been highlighted in red in the main text.

Responses to Reviewer #1 (Remarks to the Author):

[Comment 1] I have no further comments. My suggestions have been addressed.

Response: We appreciate your time and effort in reviewing our work, which has undoubtedly contributed to the improvement of our paper.

Responses to Reviewer #2 (Remarks to the Author):

[Comment 1] I am satisfied with the revisions made in the manuscript.

Response: Thank you for your feedback, and I'm glad to hear that you are satisfied with the revisions made in the manuscript. Your comments have been invaluable in enhancing the manuscript's clarity and accuracy.

[Comment 2] Given the uncertainties associated to the calculations, the title could better reflect the results by removing 'to nearly 80%'.

Response: In line with your suggestion, we removed 'to nearly 80%' from the title considering the precision of our findings. However, just stating that “*The global mean nitrogen recovery efficiency in croplands can be enhanced by optimal nutrient, crop and soil management practices*” is very trivial for those involved in the topic, since all know that an increase in NUE is possible by optimal management practices. We thus added the word “strongly”, and the revised title was listed as follows: “*Global mean nitrogen recovery efficiency in croplands can be strongly enhanced by optimal nutrient, crop and soil management practices*”.